Analysis of QTL mapping for germination and seedling response to drought stress in sunflower (Helianthus annuus L.)

Shi Huimin
Wu Yang
Yi Liuxi yiliuxivip@163.com
Hu Haibo
Su Feiyan
Wang Yanxia
Li Dandan
Hou Jianhua houjh@imau.edu.cn
Inner Mongolia Agricultural University, College of Agriculture , Huhhot , China
Kumar Sushil
Electronic publication date: 2023 May 3
Publication date: 2023
Volume: 11
Electronic Location ID: e15275
Received 2022 Oct 6; Accepted 2023 Mar 30
Copyright: ©2023 Shi et al.
Copyright year: 2023
Copyright holder: Shi et al.
License: This is an open access article distributed under the terms of the Creative Commons Attribution License, which permits unrestricted use, distribution, reproduction and adaptation in any medium and for any purpose provided that it is properly attributed. For attribution, the original author(s), title, publication source (PeerJ) and either DOI or URL of the article must be cited.
License URL: https://creativecommons.org/licenses/by/4.0/

Keywords: Sunflower, Germination and seedling stages, Drought tolerance, Single-nucleotide polymorphisms (SNPs), Quantitative trait locus (QTL), Candidate genes

Funding: The National Natural Science Fundation of China 31760396 32160450 The Special Project for Breeding Animal and Plant Varieties of Inner Mongolia Agricultural University YZGC2017004 Huimin Shi was supported by the National Natural Science Fundation of China through research grants 31760396 and 32160450. Similarly, support was given by the Special Project for Breeding Animal and Plant Varieties of Inner Mongolia Agricultural University (YZGC2017004) through research grants. The funders had no role in study design, data collection and analysis, decision to publish, or preparation of the manuscript.

==============================
Sunflower is an important oilseed crop across the world. It is considered as a moderately drought tolerant plant, however, its yield is still negatively affected by drought stress. Improving drought tolerance is of the outmost important for breeding. Although several studies have documented the relationship between the sunflower phenotype and genotype under drought stress, but relatively few studies have simultaneously investigated the molecular mechanisms of drought tolerance in the sunflower at different growth stages. In this study, we conducted quantitative trait locus (QTL) analysis for different sunflower traits during the germination and seedling stages. Eighteen phenotypic traits were evaluated under well-watered and drought stress conditions. We determined that the germination rate, germination potential, germination index, and root-to-shoot ratio can be used as effective indexes for drought tolerance selection and breeding. A total of 33 QTLs were identified on eight chromosomes (PVE: 0.016%–10.712% with LOD: 2.017–7.439). Within the confidence interval of the QTL, we identified 60 putative drought-related genes. Four genes located on chromosome 13 may function in both germination and seedling stages for drought response. Genes LOC110898128, LOC110898092, LOC110898071, and LOC110898072 were annotated as aquaporin SIP1-2-like, cytochrome P450 94C1, GABA transporter 1-like, and GABA transporter 1-like isoform X2, respectively. These genes will be used for further functional validation. This study provides insight into the molecular mechanisms of the sunflower’s in response to drought stress. At the same time, it lays a foundation for sunflower drought tolerance breeding and genetic improvement.

Introduction

The sunflower (Helianthus annuus L.2n = 34), belonging to the Compositae family and the Helianthus genus, is one of the most important oil crops in the world (De Oliveira Filho & Egea, 2021). One of its main characteristics is its wide adaptability (Dimitrijevic & Horn, 2017). However, it grows mostly in arid or semi-arid regions, making it more susceptible to environmental stress and leading to a decline in yield and quality (Hussain et al., 2018).

Drought stress is one of the most common environmental stress factors that limits plant growth and development. It has become a major challenge for agricultural researchers and plant breeders (Nezhadahmadi, Prodhan & Faruq, 2013). The effects of drought stress on crops depend on the crop growth period, plant genotype, stress intensity, and stress duration (Hazrati et al., 2017; Laxa et al., 2019).

Seed germination and seedling establishment are critical steps in the crop life cycle. Seed germination is the beginning of plant life. Successful germination is not only crucial for seedling establishment but also for crop yield (Han & Yang, 2015). Seedling establishment is the primary stage of plant adaptation to the environment and is an important prerequisite for later morphological establishment. Drought significantly affects seed germination and seedling morphology, leading to a decline in plant density and yield, and ultimately causing significant economic losses (Gad et al., 2020).

The effect of drought on sunflowers is multi-level, causing corresponding changes in the plant’s from gene expression to physiological and biochemical indicators, and ultimately its phenotype. As a result of drought stress, a number of stress-related genes are activated. As drought stress continues, closed stomata reduce photosynthesis levels (Buriro et al., 2015), cell volume shrinks (Hoekstra, Golovina & Buitink, 2001), water potential (Ghobadi et al., 2013) and membrane stability reduces (Kramer & Boyer, 1995), and the balance of reactive oxygen is disrupted (Soleimanzadeh, 2012). Drought stress also decreases plant height, leaf surface area (LSA), and leaf relative water content (LRWC) (Buriro et al., 2015; Hammad & Ali, 2014), and increases root length and root-shoot ratio (Javaid et al., 2015).

Irrigation is the most effective way to relieve drought stress on plants (Zia et al., 2021); however, it is difficult to achieve in some areas due to lack of water or irrigation facilities. Therefore, there is an urgent need to searching for drought-related genes and genetic breeding tools that can be used to cope with water-scarce environments.

As modern molecular biology techniques rapidly develop, a large number of sequence data sets and overall changes in gene expression have been reported (Liang et al., 2017). This provides an opportunity to determine the drought response genetic mechanism in sunflowers. The crop performance represents the final result of interactions between thousands of genes and environmental conditions (Collins, Tardieu & Tuberosa, 2008). Using quantitative trait loci (QTL) is an effective method to dissect complex quantitative traits that has been widely used in many crops, such as Arabidopsis (El-soda et al., 2015), millet (Pennisetum glaucum) (Yadav, Sehgal & Vadez, 2011), Chinese cabbage (Festuca pratensis Huds.), and others (Alm et al., 2011).

In sunflowers, Hervé et al. (2001) reported four QTL for chlorophyll concentration and one QTL for relative water content (RWC), which explained 53% and 9.8%, respectively, of the phenotypic variation. Al-Chaarani et al. (2005) constructed a recombinant inbred line (RIL) population with “PAC-2” and “RHA-266” as parents and constructed a linkage map with 367 markers using AFLP and SSR. Several QTL related to the following indexes were identified: time to 50% germination, percentage of germinated seeds, shoot and root length, fresh weight of shoots and roots, dry weight of shoots and roots, and percentage of normal seedlings. A total of 39 QTLs were identified (Al-Chaarani et al., 2005). Abdi et al. (2012) also used this RIL population to detect QTL and identified one to 11 QTLs in the sunflower linkage map constructed with SSR markers. Most of the QTLs overlapped on different linkage populations, which was consistent with the phenotypic correlation results (Abdi et al., 2012). Abdi et al. (2013) selected the same parents as materials and detected three and six QTLs for chlorophyll concentration and RWC, respectively, under flooded conditions. However, seven and two QTLs for chlorophyll concentration and RWC were identified under water stress conditions, respectively (Abdi et al., 2013). Haddadi et al. (2011) used 304 AFLP markers and 191 SSR markers for QTL localization of traits such as days from sowing to flowering, plant height, yield, and leaves in RIL populations under two watering conditions. Among them, three to seven QTLs were localized to each trait (Haddadi et al., 2011). Masalia et al. (2018) detected a total of 13 seedling trait-related regions using 288 sunflower association localization (SAM) populations under two water conditions. They also identified 5,302 unique genes based on the HanXRQ sunflower genome annotation v1.2 (Masalia et al., 2018). Barnhart et al. (2022) found that a large number of genes were involved in drought stress and osmotic stress; specifically, the most abundant DEGs were found under PEG stress. Additionally, the response to these stresses was mostly due to non-overlapping sets of co-expressed genes determining the differences in sunflower response to stress. However, the differences between different stresses still need to be further investigated (Barnhart et al., 2022). Wu et al. (2022a) identified 14 candidate genes through the integration of a genome-wide association study and RNA-sequencing.

Most of these previous QTL studies in sunflowers involved a small number and low density of markers. There have also been few reports about the genetic basis of drought resistance in sunflowers at the germination and seedling stages. The aim of this study was to construct a high-density genetic linkage map based on SNP molecular markers in order to: (1) analyze the phenotypic variation in the RIL population, (2) detect the QTL most closely associated with drought-related traits in sunflower germination and seedling stages under two water conditions, and (3) predict important genes associated with drought tolerance. The results of this study will provide insight into the molecular mechanism of early drought response in sunflowers, and a basis for the genetic improvement of the sunflowers.

Materials and Methods

Plant materials

We collected 226 sunflower inbred lines from Chinese provinces and other countries. A comprehensive drought tolerance coefficient value (D-value) was used to evaluate the drought tolerance of all accessions (Li et al., 2015b). The two inbred lines with the highest (K58) and lowest D-values (K55) were selected as parental materials (Zhen et al., 2021). The comparison of drought tolerance coefficients between K55 and K58 traits is shown in Fig. 1.

Figure 1 Comparison of drought tolerance coefficients between K55 and K58 traits.

Data shown as mean ± s.e.m. Student’s t-test was used to generate the P values.

Using the single-seed descent method, we constructed a recombinant inbred lines (RILs) population containing 147 F7 individuals for the construction of a genetic linkage map (Lyu et al., 2020). The F7 population and both parents were grown in the experimental field at the Inner Mongolia Agricultural University in Hohhot, China (111.71, 40.82, 1,000 m above sea level).

Seed germination test

The experiment was conducted in the summer of 2018 at the Inner Mongolia Agricultural University, China. Seeds with fully mature, healthy, and uniform sizes were sorted for drought stress experiments. First, they were disinfected with 75% absolute ethanol for 30-60s and rinsed with distilled water 3-5 times. Then, we added 20% NaClO solution to dilute and disinfect for 5 min, rinsed 3-5 times with distilled water to wash away the residual NaClO solution on the surface, and soaked in deionized water for 24 h at room temperature.

All genotypes were tested under two treatments: control (0-PEG 6000) and drought stress (18%-PEG 6000). The sterilized seeds were divided into three parts, each containing 50 seeds. We placed the seeds in a Petri dish (10 cm diameter) with a layer of sterilized filter paper placed in advance. Next, 10 ml of the 18% PEG-6000 solution was added and the same dose of distilled water was used as a control. The experiment was repeated thrice. Petri dishes were placed in a constant temperature light incubator and incubated in the dark for 3 days. On day 4, 5 mL of water was added to the control, and the incubation was started by adjusting the temperature to 25 ± 2 °C, with a light cycle of 8h/16 h (light/dark) and a light intensity of 3,000 1x. Ten days counted as a germination cycle, and the number of seeds germinated was recorded every day from 1 to 10 days.

On day 10, five seeds with consistent growth were selected from each petri dish to measure germ length (GL), radicle length (RL), germ fresh weight (GFW), and radicle fresh weight (RFW).

The radicle and germ respectively were placed in a kraft paper bag, sealed, marked, and placed in the drying box during the drying process. The seedlings were placed in an oven for 20 min at 105 °C and then dried to constant weight at 80 °C. The germ dry weight (GDW) and radicle dry weight (RDW) were calculated.

Calculation of seed germination related parameters: GR=nN×100%

(GR: germination rate, n: total number of germinated seeds, N: total number of tests) (Liu et al., 2020) GP=aN×100%

(GP: germination potential, a: the total number of germinated seeds in the first four days) GI= ∑GtDt

(GI: germination index, Gt: number of germinated seeds on day t, Dt: corresponding to the t day of germination)

Seedling pot test

After the germination experiment, the seeds were sown in plastic flowerpots (25  × 19 × 16 cm) filled with 4 kg soil (75% sand, 20% nutrient soil, and 5% vermiculite). There were 10 seeds per pot and six pots per accession. All the plants were grown in the Experimental Greenhouse at the Eastern District of Inner Mongolia Agricultural University (light/dark cycles: 14 h/10 h; 28/22 °C; 45 ± 5% relative humidity). Two groups were set up for the experiment, one for normal watering and the other for drought stress. Each group contained three pots as three biological replicates. The soil was watered thoroughly before the stress treatment to keep the soil water content in each pot consistent. Drought stress was initiated when the seedlings grew to the stage of six leaves. The water stress was carried out according to the method of Wu et al. (2022a). The drought stress (DS) soil moisture content was set to 10 ± 2%, and 30 ± 2% in the control group (CK) (Wu et al., 2022a). We measured the soil water content at 8:00 am every two days using a weight method (Soni & Abdin, 2017). The soil was supplemented with water according to the target soil moisture content. Soil water content and watering rate are shown in Table 1.

The experiment lasted for 14 days, after which five plants were randomly selected from each pot for phenotypic evaluation. The plant height of seedling (PHS) was measured directly with a ruler. Seedlings were then separated from the roots and the ground fresh weight of seedling (GFWS) and underground fresh weight of seeding (UFWS) were measured. They were dried at a constant temperature of 80 °C and the ground dry weight of seedling (GDWS) and underground dry weight of seedling (UDWS) were measured. Root shoot ratio (RSR) was measured using the gravimetric method. Total root length of seedling (RLS) and root surface area (RSA) were measured with an LA-S root scanner (Wanshen Testing Technology Co., Ltd., Hangzhou, China). The leaf relative water content (RLWC) was determined using the saturate water method according to Galmés et al. (2011).

The relative water content of the leaves was measured using the saturation weighing method and the formula is: RLWC=Wf−WdWt−Wd×100%

where Wf: fresh weight of the leaves, Wd: dry weight of the leaves, Wt: weight of the leaves after being saturated with water (Kiani et al., 2009).

Statistical analysis

The mean, variance, standard deviation, coefficient of variation, and Pearson’s correlation of phenotypic data were calculated using SPSS 23.0 software (SPSS for Windows, V23.0.0; SPSS, Chicago, Illinois; Tisné et al., 2008). The t-test was used to analyze the differences in phenotypic traits between parents. We also conducted analysis of variance (ANOVA) plot of parents and normal distribution of population phenotypic traits using GraphPad Prism (v8.0.2) software.

Construction of genetic linkage map

The linkage map used in this study was previously constructed by our laboratory (Lyu et al., 2020). The high-density genetic map consisted of 4,912 SNP and 93 SSR markers distributed on 17 linkage groups, with a total genetic distance of 2,425.05 cM. The average spacing between markers was 0.49 cM.

QTL mapping

QTL analysis was performed with MapQTL 4 (Van Ooijen et al., 2000) using the composite interval mapping method (CIM; Zeng, 1994). A threshold of log of odds ratio (LOD) ≥ 2.0 was used to declare suggestive QTL (Lander & Kruglyak, 1995). Alleles with positive- or negative-effects were derived from K58 or K55, respectively. The QTL nomenclature was designated as: q + trait abbreviation + chromosome number + QTL number, where “n” was added at the end of the QTL name to represent control conditions.

Identification of candidate genes

The physical and genetic map alignment relationship was confirmed using all markers in the QTL map. A logarithm of odds (LOD) threshold of 2.0 was chosen as evidence of the presence of QTL, and the genes located within the QTL were considered potential candidate genes (Chao et al., 2017). We used the sunflower genome released in 2017 as the reference genome (Badouin et al., 2017) and utilized data such as GO, KEGG, COG, NR, Pfam, and Swiss-Prot according to the function of candidate genes annotated in the library.

Table 1 Comparison table of soil water content and irrigation amount.

Stress time	Soil water content	Soil irrigation amount	
	Drought-stress	Control	Drought-stress	Control	
2d	8.50%	28.50%	60 ml	60 ml	
4d	9.50%	28.70%	20 ml	52 ml	
6d	8.00%	29.50%	80 ml	20 ml	
8d	8.80%	28.00%	48 ml	80 ml	
10d	8.80%	29.00%	48 ml	40 ml	
12d	8.60%	28.50%	56 ml	60 ml	
14d	8.30%	29.00%	68 ml	40 ml	

Result and Analysis

Frequency distribution test and phenotypic analysis

Our results showed the normal distribution of drought-resistance-related traits of both parents and 147 lines in the RIL population under CK and DS across two periods. All traits were continuously distributed and most of them were normally or partially normally distributed. Thus, the requirements for QTL localization were met (Figs. 2 and 3).

Under CK conditions, the difference between the average values of the traits of the two parents was not very obvious. The GLn, GFWn, GDWn, RFWn, RDWn, GRn, GPn, GIn, RSAn, RLSn, RLWCn, PHSn, GFWSn, UFWSn, GDWSn, and UDWSn in K58 were higher than K55. Only the RSRn was lower than K55. The coefficient of variation of the RIL population ranged from 7.746% to 65.905% (Table 2).

Under DS conditions, all the traits of K58 traits were higher than those of K55, and the coefficient of variation of the RIL population ranged from 16.850% to 80.633%. Compared with the corresponding values under CK growth condition, the average values of all traits under DS conditions were significantly lower in the RIL population, except for five traits (RSA, RSR, RLS, UFWS, and UDWS). This shows that drought stress inhibited the growth of sunflower shoots, but promoted the growth of roots (Table 3).

Figure 2 Frequency distributions of sunflower traits at the germination stage.

CK, under normal watering conditions; DS, under drought stress conditions.

Figure 3 Frequency distributions of sunflower traits at the seedling stage.

CK, under normal watering conditions; DS, under drought stress conditions.

Correlation analysis

Pearson’s correlations were computed for all traits using the mean values (Table S1). Under the CK conditions, all significant correlations were positive except for the correlation between GL and RL with the GDW of the germination stage, with r = 0.977**(GP and GR), r = 0.927**(GP and GI), and r = 0.950**(GR and GI), respectively. In the seedling stages, a positive and significant correlation was found between UFWS and GL, RL, GFW, GDW, RFW, RDW, RSA, RSR, RLS, and GFWS under the control conditions. Similarly, the UDWS with RL, GFW, RFW, RDW, RSR, UFWS, and GDW showed the highest positive and significant correlations. Furthermore, a positive and significant correlation was found between GDWS with PHS, and GFWS, but a negative correlation was found with RSR, with r = −0.358**. PHS had positive and significant correlations only with GL under both treatments, as well as RLS and RSA.

Under drought stress, all of the very significant correlations were positive in the germination stage, with r = 0.919**(GP and GR), r = 0.921**(GP and GI), and r = 0.956**(GR and GI), respectively. In the seedling stages, RLWC with GP, GR, and GI had a negative and significant correlation, but a positive correlation with RSR under drought stress. Similarly, the highest negative and significant correlations were observed between RSR with PHS, GFWS, and GDWS. In addition, GDWS with RLS, PHS, and GFWS showed highly significant positive correlations.

Table 2 Analysis of phenotypic traits at germination and seedling stage under normal watering conditions.

	Parents	RIL population	
	K55	K58	Minimum	Maximum	Mean ±SD	CV %	Skewness	Kurtosis	
GLn	6.1	6.12	1.493	10.08	4.847 ± 1.727	35.628	0.548	−0.138	
RLn	5.24	5.24	1.26	15.907	6.398 ± 3.036	47.454	0.4	−0.156	
GFWn	0.95	1.09	0.297	4.62	2.402 ± 0.822	34.233	−0.038	−0.095	
GDWn	0.09	0.092	0.083	0.373	0.219 ± 0.061	28.019	0.069	−0.207	
RFWn	3.78	3.79	0.017	1.267	0.457 ± 0.301	65.905	0.855	−0.014	
RDWn	0.3	0.35	0.005	0.111	0.039 ± 0.021	53.473	0.788	0.664	
GPn	82.667	94.667	14.667	98.667	74.613 ± 23.622	31.659	−1.105	0.221	
GRn	82.667	94.667	6.667	98.667	70.002 ± 26.184	37.405	−0.851	−0.472	
GIn	11.26	12.383	0.507	12.333	7.707 ± 3.247	42.131	−0.403	−0.916	
RSAn	24.013	25.907	0.5	12.12	3.272 ± 1.332	40.726	2.352	12.275	
RSRn	0.274	0.177	0.009	0.341	0.092 ± 0.054	58.365	1.228	2.29	
RLSn	25.12	26.78	10.14	87.99	22.841 ± 8.773	38.409	3.054	19.408	
RLWCn	0.917	0.95	0.69	0.99	0.868 ± 0.067	7.746	−0.675	0.038	
PHSn	25.12	26.783	11.77	29.6	17.010 ± 2.843	16.713	0.933	2.053	
GFWSn	13.81	15.31	2.48	15.9	8.239 ± 1.817	22.055	−0.207	2.045	
UFWSn	2.62	3.04	0.34	1.98	1.027 ± 0.341	33.167	0.578	−0.056	
GDWSn	2.26	3.95	0.25	2.87	0.873 ± 0.430	49.254	1.824	4.211	
UDWSn	0.62	0.7	0.027	0.573	0.085 ± 0.078	56.826	7.048	70.611	

Table 3 Analysis of phenotypic characters at germination and seedling stages under drought stress conditions.

	Parents	RIL population	
	K55	K58	Minimum	Maximum	Mean	CV (%)	Skewness	Kurtosis	
GL	3.96	4.01	0.85	2.83	1.516 ± 0.403	26.551	0.908	0.719	
RL	3.63	3.67	0.8	9.24	3.087 ± 1.962	63.571	0.995	0.35	
GFW	0.25	0.26	0.03	1.613	0.613 ± 0.363	59.24	0.549	−0.208	
GDW	0.02	0.03	0.004	0.443	0.159 ± 0.078	49.211	0.105	0.508	
RFW	0.94	1.46	0.002	0.26	0.087 ± 0.068	77.977	0.929	−0.208	
RDW	0.12	0.19	0.001	0.05	0.016 ± 0.012	76.116	1.072	0.628	
GP	29.333	54	1.333	94.667	43.811 ± 26.389	60.233	0.123	−1.181	
GR	25.333	42.667	1.001	94.667	33.391 ± 26.924	80.633	0.616	−0.833	
GI	2.383	4.223	0.057	10.277	3.338 ± 2.365	70.883	0.745	−0.36	
RSA	14.84	16.05	0.86	14.55	5.064 ± 1.791	35.362	1.573	5.704	
RSR	0.314	0.379	0.027	1.035	0.233 ± 0.124	52.941	2.273	11.409	
RLS	21.66	33.12	10.34	69.7	28.253 ± 11.635	41.184	1.013	0.842	
RLWC	0.817	0.893	0.35	0.9	0.604 ± 0.102	16.85	0.532	0.183	
PHS	21.66	23.12	9.13	24.83	14.476 ± 3.090	21.347	0.69	0.243	
GFWS	9.5	9.93	1.88	9.29	4.785 ± 1.322	27.639	0.713	0.608	
UFWS	1.13	1.42	0.29	3.04	1.171 ± 0.386	32.992	0.818	3.014	
GDWS	1.94	1.82	0.018	1.51	0.485 ± 0.208	42.902	1.548	3.914	
UDWS	0.61	0.69	0.02	0.88	0.104 ± 0.077	73.782	7.327	71.757	

QTL mapping of related traits under two water conditions at germination and seedling

Using the high-density genetic linkage map and the phenotypic data of related traits in two sunflower growth stages under two water conditions (Fig. 4, Table 4), a total of 33 QTLs were detected in the RIL populations. Among them, 15 QTLs were detected under CK and 18 QTLs were detected under DS. Six QTLs were detected on the chromosome, which was the largest number. Based on QTL detection results, three common QTL (Co-9, Co-13, and Co-17) were detected on Chr. 9, Chr. 13, and Chr. 17, respectively (Table 5).

Figure 4 QTL mapping for each trait in the sunflower germination and seedling stages.

The location of putative QTL for all traits of control water conditions and drought stress conditions. QTL for germination traits are shown in red and QTL for seedling traits are shown in green. The numbers of linkage groups are shown at the lower left corner, and the genetic distance (cM) and markers are indicated on the right. The QTL for germination and seedling traits are shown to the left of the linkage groups.

QTL mapping analysis of germination-related traits

During the germination period, a total of 16 QTLs for seven traits were detected. Among them, no QTL were detected for RL and RDW under the two water conditions. The largest number of QTLs were distributed on Chr. 9, of which three QTLs (qGPn-9-1, qGRn-9-1, qGIn-9-1) were located at the same region. Another three QTLs (qGLn-13-1, qGDW-13-1, qGI-13-1) were located on chromosome 13, and their positions were close.

Under normal watering conditions, a total of six QTLs were detected, with one QTL each for GL, RFW, GDW, GP, GR, and GI. The QTL linked to the GI in Chr. 9 had the highest LOD value (3.381). The QTL linked to the GR in Chr. 9 contributed the most, with a rate of 7.603%. No QTL were detected for RL, RDW, and GFW under control conditions.

Under drought stress conditions, a total of 10 QTLs were detected, including one QTL for GL, one QTL for RFW, three QTLs for GFW, two QTLs for GDW, one QTL for GP, and two QTLs for GI. The LOD value of the QTL long-linked to the embryo in Chr. 3 was the largest (2.71), and the phenotypic contribution rate was 7.83%. The QTL phenotype of qGL-6-1, which is located on Chr. 11, was linked to the GDW, which had the largest contribution rate of 7.913%. No QTL were detected for RL, RDW, and GP under control conditions.

QTL mapping analysis of related traits at seedling stage

A total of nine traits (RSA, RSR, RLS, RLWC, PHS, GFWS, UFWS, GDWS, and UDWS) of sunflower RILs at the seedling stage were QTL localized under normal watering and drought stress conditions, and a total of 17 QTLs were mapped. Among these, the highest number of QTL was found on Chr. 17. The physical locations of qRSA-17-1 and qRLWCn-17-1 were close, and qRSRn-13-1 and qUDWS-13-1 were located at the same physical location.

Under the control conditions, nine QTLs were detected (RSA, RSR, GLS, RLWC, PHS, GFWS, UFWS, GDWS, and UDWS), and one QTL was obtained for each indicator. The highest LOD value (5.187) of the QTL was linked to GFWS in the 10th linkage group. The phenotypic contribution rate of the QTL linked to the RSR in Chr. 13 was the largest (10.712%).

Under drought stress conditions, eight QTLs were detected, of which one QTL was obtained each for six indicators (RSA, RSR, RLS, UFWS, GDWS, and UDWS) and two were obtained for leaf relative water content. The QTL linked to the UDWS on Chr. 13 had the largest LOD value, which was 7.439. The phenotypic contribution rate of the QTL linked to the UFWS on Chr. 17 was the largest (6.261%). No QTL were detected for PHS under control conditions.

Common QTL located during germination and seedling stages

In this experiment, some QTL sites were found to be located in the same physical location or marker position. There were two pairs of sites with overlapping physical locations at the germination and emergence stages: qGDW-11-1 and qRLSn-11-1, and qGI-9-1 and qGDWS-9-1.

Table 4 QTL mapping results for each trait at germination and seedling stage.

	Trait	LGS	Start (CM)	End (CM)	Marker interval	MLOD	ADD	R2	
Germination stage	qGLn-13-1	13	110.923	111.265	Marker91222-Marker90262	2.331	0.413	5.536	
qRFWn-14-1	14	88.085	88.085	Marker101844-Marker101845	2.783	0.022	0.545	
qGDWn-15-1	15	99.46	99.46	Marker108951-Marker108950	2.067	−0.015	5.631	
qGRn-9-1	9	53.892	54.368	Marker56597-Marker56457	3.133	6.642	7.603	
qGPn-9-1	9	53.892	54.368	Marker56524-Marker56457	2.841	7.183	7.222	
qGIn-9-1	9	53.892	54.368	Marker56595-Marker56457	3.381	0.833	6.343	
qGL-3-1	3	72.358	72.844	Marker21833-Marker21837	2.71	0.116	7.83	
qRFW-15-1	15	156.522	159.578	Marker106188-Marker106186	2.192	0.005	0.6	
qGFW-3-1	3	33.613	34.157	Marker22083-Marker22471	2.278	0.079	4.287	
qGFW-6-1	6	52.738	52.738	Marker37377-Marker38038	2.107	−0.078	4.151	
qGFW-8-1	8	61.787	61.787	Marker52178-Marker52247	2.383	−0.092	5.854	
qGDW-11-1	11	96.061	96.426	Marker82816-Marker82874	2.599	−0.023	7.913	
qGDW-13-1	13	114.029	114.374	Marker91681-Marker93405	2.417	−0.022	7.439	
qGR-5-1	5	50.005	50.347	Marker29875-Marker29870	2.257	−6.711	5.457	
qGI-9-1	9	77.509	77.851	Marker55272-Marker55271	2.731	0.494	3.603	
qGI-13-1	13	107.782	108.125	contig539-Marker89965	2.699	−0.669	6.62	
Seedling stage	qRSA-17-1	17	140.495	140.838	Marker122640-ORS418_6	2.461	−0.434	5.796	
qRSR-16-1	16	112.432	112.774	Marker114321-Marker114349	2.203	−0.025	5.656	
qRLS-17-1	17	93.997	93.997	Marker134419-Marker134446	3.338	2.634	4.119	
qRLWC-5-1	5	99.445	99.445	Marker34142-Marker34110	2.152	0.009	0.768	
qRLWC-5-2	5	102.219	103.618	ORS172_1-Marker38315	2.035	0.01	0.919	
qUFWS-17-1	17	29.001	29.478	Marker122927-Marker122926	2.254	0.1	6.261	
qGDWS-9-1	9	78.893	78.893	Marker55276-Marker55287	2.745	0.055	3.369	
qUDWS-13-1	13	58.444	60.796	Marker94413-Marker92773	7.439	0.003	0.141	
qRSAn-10-1	10	104.611	104.747	Marker72149-Marker72828	2.941	−0.387	8.02	
qRSRn-13-1	13	58.444	60.796	Marker94413-Marker92773	2.917	0.018	10.712	
qRLSn-11-1	11	92.652	93.47	Marker82802-Marker82805	3.817	1.203	1.386	
qRLWCn-17-1	17	140.838	143.636	ORS418_6-Marker125509	3.017	−0.01	2.487	
qPHSn-5-1	5	22.695	23.173	Marker36571-Marker36570	4.028	0.668	4.407	
qGFWSn-10-1	10	102.325	103.008	Marker72733-Marker72148	5.187	−0.505	6.231	
qUFWSn-4-1	4	39.767	40.176	Marker29520-Marker29633	2.103	0.085	6.424	
qGDWSn-7-1	7	77.346	77.346	Marker41473-Marker45080	3.498	−0.057	1.61	
qUDWSn-9-1	9	93.466	93.808	Marker54765-Marker54588	4.305	−0.001	0.016	
Notes.

LGs, chromosome on which was located; ADD, additive effect; R2, variation accounted for by each putative QTL.

Table 5 Common QTLs controlling drought tolerance-related traits.

Common QTL	Peak range (cM)	Consensus QTL	LGs	Confidence intervals	LOD	R2	Add	
Co-9	53.892-54.368	qGRn-9-1	9	Marker56597-Marker56457	3.133	7.603	6.642	
qGPn-9-1	Marker56524-Marker56457	2.841	7.222	7.183	
qGIn-9-1	Marker56595-Marker56457	3.381	6.343	0.833	
Co-13	58.444-60.796	qUDWS-13-1	13	Marker94413-Marker92773	7.439	0.141	0.003	
qRSRn-13-1	Marker94413-Marker92773	2.917	10.712	0.018	
Co-17	140.495-143.636	qRSA-17-1	17	Marker122640-ORS418_6	2.461	5.796	−0.434	
qRLWCn-17-1	ORS418_6-Marker125509	3.017	2.487	−0.01	
Notes.

LGs, chromosome on which was located; ADD, additive effect; R2, variation accounted for by each putative QTL.

Candidate genes in response to drought stress

Based on candidate genes annotated in the database function, we found that Clusters of Orthologous Groups (COG) database were mainly annotated to the following: two were involved in signal transduction mechanisms; two were involved in defense mechanisms; three were involved in posttranslational modification, protein turnover, chaperones; six were involved in carbohydrate transport and metabolism; one was involved in inorganic transport, metabolism, transport, and catabolism; and one gene was involved with translation, ribosomal structure, and biogenesis under the two water conditions. A total of 60 candidate genes were annotated and screened for oil sunflowers across two growth periods under two environments. The candidate genes were located within 17 QTLs on nine linkage groups (3, 4, 6, 7, 8, 9, 10, 13, and 14) and associated with 14 individual traits (GFW, GDW, RFW, GR, GP, GI, RSA, RSR, GFWS, UFWS, GDWS, and UDWS), excluding GL, RL, RDW, RLWC, RLS, and PHS. Four candidate genes (LOC110898128, LOC110898092, LOC110898071, and LOC110898072) were jointly annotated to the germination and seedling stages, all located on Chr. 13 in the qGDW-13-1 and qRSRn-13-1 intervals, respectively (Table S2).

Candidate genes in response to drought stress at the germination stage

Based on our gene function annotation, some candidate genes were screened for their coping mechanism under drought stress at the germination stage. LOC110877496 was within three QTLs (qGRn-9-1, qGPn-9-1, and qGIn-9-1), associated with GR, GP, and GI under control conditions, respectively, which encoded WRKY transcription factor 32. LOC110877508 was within three QTLs (qGRn-9-1, qGPn-9-1, and qGIn-9-1), associated with GR, GP, and GI under control conditions, respectively, encoding NADPH–cytochrome P450 reductase. Two genes, LOC110902688 and LOC110897853, on Chr. 13 were associated with GDW under drought stress, which was annotated as glutathione S-transferase T1-like, and they were involved in posttranslational modification, protein turnover, and chaperones. LOC110902670 and LOC110902674 were within qGDW-13-1, associated with GDW under drought stress, and encoded late embryogenesis abundant protein At1g64065-like.

Candidate genes in response to drought stress at the seedling stage

Based on our gene function annotation, many candidate genes were screened for their coping mechanism under drought stress at the seedling stage. LOC110886051 in both QTLs qRSAn-10-1 and qGFWSn-10-1 was associated with RSA and GFWS under control conditions, and encoded dehydrin LEA. LOC110886021 was detected in two QTLs (qRSAn-10-1 and qGFWSn-10-1) associated with RSA and GFWS under control conditions, and encoded WRKY transcription factor 6-like. Three genes (LOC110868317, LOC110868358, and LOC110868335) were located in qGDWSn-7-1 associated with GDWS under control conditions and encoded for aquaporin PIP2-4-like. They were involved in carbohydrate transport and metabolism.

The gene LOC110898429 was located on Chr. 13 within the QTL qRSRn-13-1 associated with RSR under control conditions, which functioned as ABC transporter G family member 15-like. In addition, it was found to play a role in defense mechanisms. LOC110898478 was located on Chr. 13 within both QTLs (qRSRn-13-1 and qUDWS-13-1) associated with RSR under control conditions and UDWS under drought conditions, which encoded EREBP-like factor and was found to be down-regulated after drought stress. LOC11089846 6 was located within qRSRn-13-1 and qUDWS-13-1 and encoded for aquaporin TIP4-1. This gene was involved in carbohydrate transport and metabolism. LOC110898325 was located in both QTLs (qRSRn-13-1 and qUDWS-13-1) and encoded for cytochrome P450. LOC110898141 and LOC110898323 were also located in both QTLs, which encoded NAC domain-containing protein 100-like and serine/threonine-protein kinase PBL19.

Common candidate genes in response to drought stress at the germination and seedling stages

Based on the annotation information, some common candidate genes related to drought resistance were found in two periods. LOC110898092 which was located on Chr. 13 within the three QTLs (qGDW-13-1, qRSRn-13-1, and qUDWS-13-1) associated with GDW, UDWS under drought conditions, and RSR under control conditions, which encoded cytochrome P450. LOC110898128 was located on Chr. 13 within the two QTLs (qGDW-13-1 and qRSRn-13-1) associated with GDW under drought, and RSR under control conditions, which encoded aquaporin SIP1-2-like. LOC110898071 and LOC110898072 were located on Chr. 13 within both QTLs (qGDW-13-1 and qUDWS-13-1), which encoded GABA transporter. These important candidate genes provided insight into drought tolerance in Helianthus annuus L, and their function could be verified further using a trans-gene method in the future.

Discussion

PV under drought stress

Since the drought resistance of crops is a comprehensive trait, a single index is often not representative of overall performance. To evaluate the drought resistance of plants, there are different indicators. However, each indicator may only involve a single biological process. In crops, the drought resistance characteristics related to the final production (product) are a series of biological processes controlled by a series of genetic processes (Kacperska, 2004). All of the biological processes of a single DT will eventually be reflected in the growth of the plant and its final product (Bota, Medrano & Flexas, 2004). Therefore, the only way to measure the drought resistance of crops is to select multiple indicators. The drought resistance of crops can be evaluated only by developing as many indicators as possible in the variety selection process. In this study, many morphological indicators were used to identify QTL related to sunflower drought resistance. Under drought stress, 17 indicators of all traits (GL, RL, GFWS, UFWS, and UDWS; excluding GDWS, GFWS, GDW, RFW, RDW, GP, GR, GI, RSA, RSR, RLS, RLWC, and PHS) were higher in K58 than in K55 at both developmental stages, indicating that K58 is more drought-resistant (Table 3). This indirectly indicates the reliability of the materials selected in this study.

In this study, the drought tolerance of sunflowers was evaluated based on 18 different traits, representing the germination and seedling periods. All 18 traits are directly related to drought stress. Table 3 shows that under the two water conditions, most of the traits in the germination period had very significant positive correlation, among which the GR, GP, and GI had the largest correlation. Under drought stress, the GR, GP, and GI were all significantly positively correlated with other seedling stage indicators. Except for GL, the correlation coefficients were all larger than those under control conditions. This shows that the GR, GP, and GI had the closest correlation with other indicators, and they can be used as important indicators for the evaluation of drought resistance during the germination period.

Correlation analysis of the two-period traits and both treatments (Table S1) revealed that under drought stress, the highest negative and significant correlations were observed between RSR with PHS and GFWS. However, RSR was unrelated to PHS and GFWS under control conditions. In addition, under control conditions, a correlation was not found between RLWC and RSA. However, RLWC with RSA showed significant positive correlations under drought stress. These results indicate that on one hand, drought promotes the expansion of sunflower roots and increases water absorption. On the other hand, it also hinders the growth of the upper part of the ground and reduces water consumption and loss. Studies have confirmed that wheat RSR increases under drought stress (Liu, Li & Xu, 2004), and root-to-shoot ratio increases significantly under severe drought stress (Zhou et al., 2018). The RSR had a significant negative correlation with the relative soil water content. The water absorbed by the root system is used for its growth and development. Drought significantly modified root morphological traits and increased root mortality, and the drought-induced decrease in root biomass was less than in shoot biomass, causing higher RSR (Zhou et al., 2018). In summary, during the early stage of plant growth and development, the RSR can be used as an important index to evaluate the drought resistance of sunflower seedlings.

Genetic basis and QTL of sunflower drought resistance

In recent decades, QTL mapping has been one of the most important methods used to mine quantitative trait loci in plants, and a variety of populations with different genetic structures have been developed for mapping. RIL is a permanent segregating population, so it is often used as a QTL localization population (McCouch & Yano, 1997). In this study, the RIL population was phenotypically analyzed, and QTL was mapped for traits associated with two stages of sunflower growth and development (germination and seedling stage) under two moisture conditions. A total of 33 QTLs regions were detected on 14 chromosomes of the sunflower. The QTL cluster of three related traits was located in the same physical location or adjacent location in the same linkage group: Co-9, Co-13, and Co-17, respectively (Table 5). In addition, qGI-9-1 and qGDWS-9-1 had similar positions, the distance between qGDW-11-1 and qRLSn11-1 was very close. especially in Chr. 9, and the distance between the two sites on the linkage group was close to 1 cM. Although these QTL control different traits, and their contributions are different. The Co-location of QTL for different traits may mean that there is a multi-effect or tight linkage between QTL for controlling traits (Tuberosa et al., 2002). These results could also be seen from the correlation analysis of each trait and were consistent with the results obtained by Abdi et al. (2013) using the RIL population under water stress (Abdi et al., 2013). These markers had pleiotropic effects that had a very important impact on the marker-assisted selection and identification of candidate genes (Schrooten & Bovenhuis, 2002). Therefore, we focused on those that were associated with more than one trait.

Previous studies (Malosetti et al., 2008; Messmer et al., 2009) indicated that it was inaccurate to value and identify the effect of QTL in a single environment due to the instability of different environments. Therefore, a QTL is considered a stable QTL when it can be detected in different environments. Co-13 and Co-17, as well as qGLn-13-1, qGDW-13-1, and qGI-13-1, were detected in different environments, so they are considered stable QTL sites. Kiani et al. (2009) detected the QTL that controlled the relative water content of sunflower leaves on Chr. 5 and 17 under two water conditions, respectively, located on the same chromosome as this study. Interestingly, Wu et al. (2022a) found that the SNP locus S5_195713371 was located on chromosome 5 and closely associated with the relative water content of leaves, and this was coincidentally located in the qRLWC-5-1 region in our study. However, it was not at the same loci as this study, which may have been caused by the different densities of the mapping. In addition, several QTL associated with important drought tolerance traits were detected on Chr. 13. Thus, Chr. 13 is of particular importance in the drought resistance of sunflowers. These stable QTL deserve priority in future research using fine mapping, candidate gene identification, and molecular mechanism analysis of sunflower development. Moreover, these stable QTL have the potential to improve sunflower tolerance through molecular marker assisted selection breeding.

Candidate genes in response to drought stress

Selecting important candidate genes from a large number of candidates is often a daunting task. However, integrating QTL information and gene expression variants is a common strategy for candidate gene screening (Lin et al., 2019). In this study, 60 genes were identified as candidate genes within 33 QTLs, and we suggest that these genes may be closely related to drought stress.

AQP gene

Candidate gene LOC110898128 in QTL qGDW-13-1 and qRSRn-13-1 was annotated as aquaporin SIP1;2. It has been reported that Arabidopsis has three members in the SIP subfamily (SIP1;1, SIP1;2, and SIP2;1; Johanson et al., 2001), maize has three (Chaumont et al., 2001), and rice has two (OsSIP1;1 and OsSIP2;1; Sakurai et al., 2005). A study revealed that all three members of SIP are localized to the ER membrane and expressed in a cell-specific manner in Arabidopsis thaliana, and SIP1;1 and SIP1;2 were found to have water transport activity, while SIP2;1 did not (Maeshima & Ishikawa, 2008). Venkatesh, Yu & Park (2013) reached a similar conclusion in their study of tetraploid potatoes. Ishikawa et al. (2005) found that SIP1;2 was expressed in the cotyledon and hydathode tissue of rosette leaves. Subsequently, Gururani et al. (2018) found that under PEG-induced osmotic stress, the constitutive expression of SIP1;2 induced significant changes in the photosynthetic machinery of Arabidopsis. However, none of the aforementioned studies showed any direct relationship between this gene and drought stress. Although a large number of studies have confirmed that overexpression of aquaporin genes in transgenic plants can enhance tolerance to drought stress (Lu et al., 2018; Wang et al., 2009; Xu et al., 2014), other studies have also found contrary conclusions (Jang et al., 2007; Li et al., 2015a). The different results on the relationship between plant drought resistance and AQP gene expression suggest that the physiological process of AQP regulation of plant drought resistance may have specificity. Most of the current studies on water channel proteins in drought and salt stress are based on PIPs. Whether other subtypes (such as SIP) are involved in drought and salt stress remains to be further explored. Therefore, the physiological roles of the ER membrane aquaporins in living organisms also needs to be further studied.

WRKY transcription factor (TF)

WRKY is a transcription factor (TF) known to play a role in the abscisic acid (ABA) and jasmonic acid (JA) signaling pathways in response to drought stress (Aziz et al., 2020). In this study, candidate gene LOC110877496 was located in QTL qGRn-9-1, GPn-9-1, and GIn-9-1, which were associated with GR, GP, GI under control conditions, respectively, and annotated as transcription factors WRKY32. Niu et al. (2015) found that under salinity and drought stress, WRKY32 in Kenaf (Hibiscus cannabinus) increased 6.5-fold and 7.5-fold at 12 h and 24 h, respectively. Subsequent studies found that VbWRKY32 was a positive regulator that upregulates the transcriptional level of cold response genes, increases antioxidant activity, maintains membrane stability, and enhances osmotic regulation ability, thereby improving the survival ability under cold stress (Wang et al., 2020). In summary, WRKY32 may play a role in various stresses and give plants the ability to cross-adapt. Therefore, we suspect that the WRKY32 TF plays an important role in the drought process, but its specific regulatory mechanism in drought highlights that further research is required.

Cytochrome P450

LOC110898092 located on QTL qGDW-13-1, qRSRn-13-1, and qUDWS-13-1 was annotated as cytochrome P450 94C1. LOC110877508 located on qGRn-9-1, GPn-9-1, and GIn-9-1 was annotated as NADPH–cytochrome P450 reductase-like. Studies have shown that CYP707A1 (ABA 8′-hydroxylase genes),CYP94C1 and CYP94B3 in plants under drought stress to be significantly up-regulated in leaf and root, indicating that CYP450s genes may be induced by drought stress. Under drought stress, Arabidopsis cytochrome P450 and CYP94C1 were involved in JA-Ile oxidation. The enzyme catalyzed the catabolic turnover of JA-Ile. CYP94C1 and CYP94B3 catalyzed successive oxidation steps in JA-Ile turnover (Rabara et al., 2015). After drought the JA and JA-Ile content in the Arabidopsis thaliana L. root system both increased, and the study confirmed that signal transduction between water stress-induced early JA-Ile accumulation and COI1 was necessary for full induction of the ABA biosynthesis pathway and subsequent hormone accumulation in roots of Arabidopsis plants (De Ollas, Arbona & Gómez-Cadenas, 2015). Therefore, we speculated that P450 94C1 plays an important role in drought stress.

GABA transporter

LOC110898071 and LOC110898072 are located on qGDW-13-1, and qUDWS-13-1 is an annotated GABA transporter. Studies have found that GABA is not only related to seed germination, but also plays an important role in plant drought resistance. Above all, GABA is involved in seed germination and primary and adventitious root growth. GABA levels increase during the germination of soybean (Matsuyama et al., 2009), oats (Xu et al., 2010), rice (Zhao et al., 2017), and wheat (Al-Quraan, Al-Ajlouni & Obedat, 2019; Kim, Kwak & Kim, 2018). GABA activates α-amylase gene expression and promotes seed starch degradation in a dose-dependent pattern in seed germination. More importantly, GABA acts as a downstream signaling molecule of stress-related transcription factors, such as WRKY, MYB, and bZIP, and confers drought tolerance in plants (Li, Peng & Huang, 2018). A high level of GABA also increases chlorophyll content, osmoregulation (i.e., soluble sugars and proline), and antioxidant enzyme activity in black cumin subjected to a water deficit (Rezaei-Chiyaneh et al., 2018). GABA enhancement of drought tolerance is associated with the improvement of nitrogen recycling, protection of photosystem II, and mitigation of drought-depressed plants (Li et al., 2019). Furthermore, Xu et al. (2021) revealed that guard cell GABA production in Arabidopsis was necessary and sufficient to reduce stomatal opening and transpiration water loss. It improved water use efficiency and drought resistance through negative regulation of stomatal guard cell tonoplast-localized anion transporters. They also found that this regulation of stomata occurs in both monocotyledonous and dicotyledonous plants, and that GABA does not initiate changes in the stomatal aperture. Instead, it antagonizes stomatal aperture changes, which distinguishes it from many of the signals known to regulate stomatal aperture (Hetherington & Woodward, 2003; Kim et al., 2010; Sussmilch et al., 2019). In a similar and recent study, Wu et al. (2022b) found that GABA was significantly enriched in sunflowers at different time points of drought, indicating the important role of endogenous GABA in drought response in sunflowers. However, their specific roles in early drought regulatory mechanisms in sunflowers need to be further investigated.

Conclusion

To illustrate the complexity of drought tolerance mechanisms in sunflowers and identify the genes behind these mechanisms, we used a number of genetic and genomic techniques. In this study, some important drought-related indicators were used as QTL mapping and gene identification for drought tolerance traits in sunflowers at the germination and seedling stages. Based on QTL analysis, 33 QTLs associated with drought tolerance and three unique QTL were identified. Chr. 9 and 13 were the focus of further studies. In addition, 60 drought tolerance-related candidate genes were identified, four of which were found together in the germination and seedling stages. Our results suggest that specific regions on Chr. 13 are important for improving drought tolerance in sunflowers due to the presence of many QTL associated with drought tolerance. However, further studies are needed to confirm the candidate genes associated with drought tolerance in sunflowers.

Supplemental Information

Supplemental Information 1 Raw data for plant phenotypic traits, drought tolerance, and qualitative trait mapping (Tables 2–5 and Figures 1-4)

Click here for additional data file.

Supplemental Information 2 Phenotypic correlation between traits measured under normal watering control and drought condition

** Correlation is significant at the 0.01 level. * Correlation is significant at the 0.05 level. The upper right corner of the table represents the correlation coefficients between traits under drought stress; the lower left corner of the table represents the correlation coefficients between traits under normal conditions.

Click here for additional data file.

Supplemental Information 3 A total of 60 candidate genes were screened from a RIL population at two periods

Click here for additional data file.

Additional Information and Declarations

Competing Interests

Author Contributions

Data Availability

The authors declare there are no competing interests.

Huimin Shi conceived and designed the experiments, performed the experiments, analyzed the data, prepared figures and/or tables, authored or reviewed drafts of the article, formal analysis and investigation, and approved the final draft.

Yang Wu performed the experiments, authored or reviewed drafts of the article, and approved the final draft.

Liuxi Yi conceived and designed the experiments, authored or reviewed drafts of the article, supervisior, and approved the final draft.

Haibo Hu performed the experiments, prepared figures and/or tables, and approved the final draft.

Feiyan Su performed the experiments, authored or reviewed drafts of the article, and approved the final draft.

Yanxia Wang performed the experiments, prepared figures and/or tables, and approved the final draft.

Dandan Li performed the experiments, prepared figures and/or tables, and approved the final draft.

Jianhua Hou conceived and designed the experiments, authored or reviewed drafts of the article, supervisior, and approved the final draft.

The following information was supplied regarding data availability:

The raw data is available in the Supplemental Files.

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
