# Peer review of "Analysis of QTL mapping for germination and seedling response to drought stress in sunflower (Helianthus annuus L.)"

_PeerJ, doi:10.7717/peerj.15275_

## Round 0.1 · original submission · Minor Revisions

Please revise the article as per the reviewers' comments. Especially improve the English language throughout the article.

·

Basic reporting

The manuscript by Huimin Shi et al. pointed the genes LOC110898128 (AQP Gene), LOC110877496 (WRKY TF), LOC110898092 (Cytochrome P450), LOC110898071 and LOC110898072 (GABA transporter) as candidate genes for drought tolerance in early development stages in sunflower. The results can be useful for geneticists working on sunflower breeding. The experiment was designed in logic. Data collection and interpretation were well performed including the evaluation of several traits at the same time (e GLn, GFWn, GDWn, RFWn, RDWn, GRn, GPn, GIn, RSAn, 206 RLSn, RLWCn, PHSn, GFWSn, UFWSn, GDWSn, and UDWSn).
The article is written in clear English, unambiguous and technically correct. My suggestion is to check the correct use of the word ‘most’ in all the manuscript, but in particular in the lines 240, 262. Also, to check the use of uppercase in the lines 160, 347, 447.
A few more comments as mentioned below:
- Line 238, I would like to see two or three lines explaining why you are choosing to evaluate the QTLs under CK. E.g.: What is the purpose of this analysis?
- Line 239, what is chain group No9? Do you mean chromosome 9?
- Line 240, ‘three unique QTL’ all the QTLs are unique, because by definition a QTL is a specific genome region. Maybe the word unique is not well used in this case. Consider changing the word.
- Line 247, are the QTLs overlapping? Is that why you say that they are similar?
- Line 353, maybe goes a point instead of a comma?

There is sufficient field background/context provided in the text but in lines 78-93 the most recent citation is from 2013. I am wondering if there are more current works on this topic. I also recommend reading the following papers:
-Multiple genomic regions influence root morphology and seedling growth in cultivated sunflower (Helianthus annuus L.) under well-watered and water-limited conditions
-Phenotypic and transcriptomic responses of cultivated sunflower seedlings (Helianthus annuus L.) to four abiotic stresses

The manuscript has the structure of a professional article and the raw data is shared. Besides, it is a self-contained paper with relevant results to hypotheses.

Experimental design

no comment

Validity of the findings

no comment

Additional comments

no comment

·

Basic reporting

Minor grammatical mistakes were observed across the text. These mistakes need to be rectified before final publication.

Experimental design

no comment

Validity of the findings

no comment

Additional comments

1. Line number 31 to 33: language need to recast

2. Line number 31 to 33: language of sentence need to recast " while in in water.......identified"

3. Line number 94: instead of analysis, analyse word should be used.

4. Line number 121: instead of "The group three repeated" "The experiment was repeated thrice" should be incorporated
5. Line number 199, 202: Instead of "ranges" "ranged " should be incorporated

other grammatical mistakes also need to be rectified before final publication

---

## Round 0.2 · Major Revisions

Dear author, before we can accept your revised article, there are numerous mistakes on the grammar side. Hence, you must revise the article for language improvement. Submit the clean and track change version along with proof of editing such as a certificate from an editing service.

Please try and submit your revision in the next 10 days for further consideration.

---

## Round 0.3 · accepted · Accept

Author satisfied the queries hence article accepted for publication.

·

Basic reporting

No

Experimental design

No

Validity of the findings

No

Additional comments

Author performed experiment in systematic manner, manuscript is adding information to scientific community. Language of paper, material method, result & discussion and supporting information are appropriate. I will recommend this manuscript entitled "Analysis of QTL mapping for germination and seedling response to drought stress in sunflower (Helianthus annuus L.)" for publication.